# Detection of Additives and Chemical Contaminants in Turmeric Powder Using FT-IR Spectroscopy

**DOI:** 10.3390/foods8050143

**Published:** 2019-04-26

**Authors:** Sagar Dhakal, Walter F. Schmidt, Moon Kim, Xiuying Tang, Yankun Peng, Kuanglin Chao

**Affiliations:** 1United States Department of Agriculture/Agricultural Research Service, Environmental Microbial and Food Safety Laboratory, Bldg. 303, Beltsville Agricultural Research Center, 10300 Baltimore Ave., Beltsville, MD 20705-2350, USA; sagar.dhakal@ars.usda.gov (S.D.); walter.schmidt@ars.usda.gov (W.F.S.); moon.kim@ars.usda.gov (M.K.); 2China Agricultural University, National R&D Centre for Agro-Processing Equipments, Beijing 100083, China; txying@cau.edu.cn (X.T.); ypeng@cau.edu.cn (Y.P.)

**Keywords:** FT-IR, Sudan dye, white turmeric, turmeric, curcumin, adulteration, food safety

## Abstract

Yellow turmeric (*Curcuma longa*) is widely used for culinary and medicinal purposes, and as a dietary supplement. Due to the commercial popularity of *C. longa*, economic adulteration and contamination with botanical additives and chemical substances has increased. This study used FT-IR spectroscopy for identifying and estimating white turmeric (*Curcuma zedoaria*), and Sudan Red G dye mixed with yellow turmeric powder. Fifty replicates of yellow turmeric—Sudan Red mixed samples (1%, 5%, 10%, 15%, 20%, 25% Sudan Red, *w*/*w*) and fifty replicates of yellow turmeric—white turmeric mixed samples (10%, 20%, 30%, 40%, 50% white turmeric, *w*/*w*) were prepared. The IR spectra of the pure compounds and mixtures were analyzed. The 748 cm^−1^ Sudan Red peak and the 1078 cm^−1^ white turmeric peak were used as spectral fingerprints. A partial least square regression (PLSR) model was developed for each mixture type to estimate adulteration concentrations. The coefficient of determination (R^2^_v_) for the Sudan Red mixture model was 0.97 with a root mean square error of prediction (RMSEP) equal to 1.3%. R^2^_v_ and RMSEP for the white turmeric model were 0.95 and 3.0%, respectively. Our results indicate that the method developed in this study can be used to identify and quantify yellow turmeric powder adulteration.

## 1. Introduction

Turmeric root (*Curcuma longa*) is widely used for culinary, medicinal, and cosmetic purposes and as a dietary supplement [1,2]. It exhibits anti-inflammatory, anti-cancer, antioxidant, and wound-healing properties, and is a potential therapy for Alzheimer’s disease [3,4,5,6,7]. The vibrant yellow color and medicinal value of turmeric is mainly due to the “curcumin” (diferuloylmethane) content, which varies between 0.3% to 8.6% [3,8,9,10,11]. Factors such as the soil type, cultivar, and crop fertilization affect the curcumin content in turmeric root [12,13,14,15]. 

In 2010, turmeric ranked number four among popularly sold plant products in the United States [16]. In 2016, the global market for curcumin was over $44 million, with an estimated compound annual growth rate of 13.3% [17]. With its strong commercial popularity, incidences of turmeric adulteration by different botanical products and chemical dyes have increased. Although there is no report of contamination in whole, dry, or fresh turmeric, turmeric powder has been adulterated with other powders. For example, another plant in the same genus, *Curcuma zedoaria* (white turmeric), may be mixed with *C. longa* powder [18,19]. *Curcuma zedoaria* also grows in *C. longa* producing areas. It has a lower curcumin content, but also a lower cost, higher yield per acre, and greater availability. Therefore, this has increased economic adulteration in *C. longa* [20]. Azo compounds such as Sudan dye and metanil yellow may be added to turmeric and other spice powders to improve the color [2,21]. Azo dye are organic compounds bearing the coloring azo function –N=N and aryl (or alkyl) group. The azo color dyes are commonly used to treat textiles, leather articles, and used as shoe polishes. Under anaerobic condition, these dyes reduce to aromatic amines, which are considered toxic and potentially carcinogenic. Consequently, Food Standard Agencies in the European Union (EU) have considered these synthetic compounds as illegal food additives for human consumption and banned for use in foodstuffs [21]. However, spice powders and food products contaminated with these dyes have been detected in some European Union countries, and have become a threat in developing countries where non-branded foodstuffs and spices are sold loose in unregulated marketplaces [22]. Since the azo dyes are added in the spice powder to enhance the visual appearance, a relatively high concentration (e.g., ≥1g L^−1^ Sudan) is needed to make a visual impact on a spice powder [21]. 

Different analytical methods have been applied for the detection of additives and chemical contaminants in yellow turmeric and other spice powders. High-performance liquid chromatography (HPLC), polymerase chain reaction, high performance capillary electrophoresis, and HPLC-electrospray ionization tandem mass spectrometry are commonly used techniques for authentication of spice powders [18,23,24,25]. These methods require skilled operators, and some techniques such as HPLC require fresh samples. Furthermore, a significant change in the composition of the same product from different manufacturers results in low sensitivity for chromatographic methods [26]. Parvathy et al., (2015) [26] used a DNA barcoding method to detect plant-based adulterants in turmeric powder. Although the DNA barcoding method can identify various additives, the method is purely qualitative, and requires matching the sequence generated with a database of potential biological adulterants. Di Anibal et al., (2011) [27] used high-resolution ^1^H nuclear magnetic resonance spectrometry (NMR) to detect five types of Sudan dye in different commercially available spices including turmeric. Due to the large dataset, for which only a few variables contain useful information, the NMR technique requires complex spectral processing to fully identify adulterants [27]. 

Optical sensing methods are gaining importance in food safety and quality detection due to their simplicity. Dhakal et al., (2016) [2] used Fourier Transfer-Infrared (FT-IR) and FT-Raman spectroscopic systems for qualitative detection of metanil yellow contamination in turmeric powder. In the study, the vibrational modes of turmeric and metanil yellow were identified and the structural components were assigned to the respective vibrational frequencies. The highest spectral peak of metanil yellow at 1140 cm^−1^ was selected to detect metanil yellow in the mixture samples. The study showed that the spectral peak intensity count of metanil yellow is proportional to its concentration in the range of 5% to 30%. Since turmeric is nearly insoluble in water, the spectra of the mixture samples cannot be used for quantitative modeling. Di Anibal et al., (2009) [21] used UV-visible spectroscopy coupled with multivariate classification to divide spice samples into pure and adulterated groups. Similarly, Sudan I and a blend of Sudan I and Sudan IV were identified in paprika and aji molido using UV-visible spectroscopy [28]. In a recent study, FT-Mid Infrared spectroscopy was used to identify Sudan I and IV (1% concentration each), lead chromate, lead oxide (3% each), silicon dioxide (5%), or polyvinyl chloride and gum arabic (10% each) mixed into paprika [29]. Although these spectroscopic techniques could identify contaminants, and classify pure/adulterated spice powders with high accuracy, the studies did not develop quantitative models to estimate contaminant concentrations. With powder sampling, samples are ground to a fine powder to minimize the effect of particle size on spectral quality [30]. Factors such as mixing time and the size, shape, and density of particles included influencing the mixing quality [31]. Powder samples are not suitable for quantitative analysis due to the difficulty for obtaining homogeneity. 

In our previous study, we used FT-IR spectroscopy for qualitative detection of metanil yellow contamination in turmeric powder [2]. Commercial yellow turmeric powder has also been adulterated with botanical additives. The PCR-based method [18] and DNA barcoding method [26] are reported for qualitative analysis of different brands of commercially available yellow turmeric powder. The result by DNA method showed that, among the 10-commercial brands of turmeric samples used in the study, one brand was adulterated with white turmeric [26]. The PCR analysis method used three commercial turmeric samples, and reported all the samples contained white turmeric [18]. In our current study, we developed a simple method using FT-IR spectroscopy to develop quantification models to predict the concentrations of chemical contaminant (i.e. Sudan Red G) and botanical additive (i.e., white turmeric) in commercial yellow turmeric powder. Yellow turmeric and adulterants at six concentration levels were dissolved in an organic solvent and dried to obtain well-mixed samples. Fifty replicate spectral measurements were obtained for each concentration, and partial least squares regression (PLSR) models were developed to estimate the adulterant concentrations. Our primary objectives were:(1)Identify the vibrational modes of Sudan Red G, white turmeric, and yellow turmeric.(2)Prepare yellow turmeric—Sudan Red and yellow turmeric—white turmeric mixture samples at different concentrations and acquire IR spectra.(3)Develop PLSR models to estimate concentrations of Sudan Red and white turmeric in the mixed samples.

## 2. Materials and Methods 

### 2.1. Sample Preparation

Sudan Red G (Aldrich, Carson City, NV, USA), organic yellow turmeric powder (Frontier Natural Products CO-OP, Norway, IA, USA), white turmeric powder (obtained from local market), and methanol (Aldrich, Carson City, NV, USA) were utilized. In addition, 100 mg yellow turmeric-Sudan Red mixture at 1%, 5%, 10%, 15%, 20%, and 25% Sudan Red concentration (*w*/*w*), and 100 mg yellow turmeric-white turmeric at 1%, 10%, 20%, 30%, 40%, and 50% white turmeric concentration (*w*/*w*) were prepared. Each mixture was put into a conical tube containing 5 mL methanol and vortexed for 5 min. The sample was pipetted into a petri dish immediately after mixing, and placed in a hood for 3 h to evaporate the methanol, which is followed by oven drying at 75 °C for 15 min. The process was repeated to create 50 replicate samples at each concentration level. 

### 2.2. Spectral Acquisition

A Nicolet 6700 FT-IR instrument with the OMNIC 8.1 software package (Thermo Scientific, Madison, WI, USA, spectral ranged from 4000 to 400 cm^−1^) was used for spectral measurement. The FT-IR module consists of a germanium crystal attenuated total reflectance (ATR) device, deuterated triglycine sulfate (DTGS) detector, and a KBr beam splitter. A background spectrum was collected from the empty crystal to correct the sample spectrum. The background contains peaks from dissolved gases, H_2_O, and CO_2_ present in air. To obtain the spectra, small amounts of the dried sample were placed on the ATR crystal. An average of 32 successive scans at 2 cm^−1^ intervals was collected. A single spectrum was obtained from each sample, which totaled 300 spectra for yellow turmeric—Sudan mixture samples (50 replicates × 6 concentrations), and 300 spectra for yellow turmeric—white turmeric mixture samples.

### 2.3. Data Analysis

Yellow turmeric—Sudan Red spectra in the region 400 to 700 cm^−1^ were predominantly background noise, and the region of 1700 to 4000 cm^−1^ did not contain any useful information. Therefore, only the range of 700 to 1700 cm^−1^ was selected for analysis. Similarly, yellow turmeric—white turmeric spectra in the range of 900 to 1700 cm^−1^ were selected.

Some peaks from yellow turmeric, Sudan Red, and white turmeric were not resolved in the spectra from the mixtures. With IR measurements, it is not uncommon for unknown vibrational modes to be superimposed [32]. The superposition makes it difficult to correctly identify the sample components. Fourier self-deconvolution (FSD) is a proven method to deconvolute IR spectra, which we applied to the data [33,34]. Prior to FSD analysis, the baseline was subtracted from each spectrum. The pre-processing was performed in Origin 2017 (Origin Lab Corporation, Northampton, MA, USA).

If E(*v*) and I(*x*) are spectrum and an interferogram, and ℱ and ℱ
^−1^ are the Fourier and inverse Fourier transfer in Equation (1) and Equation (2), then Fourier self-deconvolution involves multiplication of the interferogram (ℱ
^−1^ {E (*v*)}) by an apodization function (D_g_(*x*)) divided by the inverse Fourier transfer of the intrinsic line function (ℱ
^−1^ {E_o_(*v*)}) of the spectrum E(*v*) [35]. The self-deconvoluted spectrum E’(*v*) is the Fourier transform of the new interferogram I’(*x*).
(1)$$E\left(v\right)=\int_{-\infty }^{+\infty }I\left(x\right)\exp\left(i2\pi vx\right)dx=ℱ\left\{I\left(x\right)\right\},$$
(2)$$I\left(x\right)=\int_{-\infty }^{+\infty }E\left(v\right)\exp\left(-i2\pi vx\right)dv={ℱ}^{-1}\left\{E\left(v\right)\right\},$$

In FSD, the signal-to-noise ratio (SNA) of the original spectra is an important factor. Higher SNA allows deconvolution of the spectrum into individual components. Line shape width is another factor that influences deconvolved spectral quality. Higher line shape width values result in narrower deconvolved peaks and vice versa. Based on a trial-and-error method, a line shape width of 20 was selected to deconvolve the spectrum at all concentrations. After FSD, the spectra were smoothed for further analysis. 

After Fourier self-deconvolution, the spectra were randomly divided into calibration (60%) and validation sets (40%). PLSR models (Matlab R2013a, MathWorks, Natick, MA, USA) were developed to estimate the extent of contamination in the mixture samples. Furthermore, 180 spectra were utilized to develop the concentration prediction models and 120 spectra were used to validate the models. Coefficient of determination (R^2^) and root mean square error (RMSE) values were computed to evaluate model performance.

## 3. Results and Discussion

### 3.1. Assignment of Spectral Bands

Figure 1a shows the IR spectra of white turmeric, yellow turmeric, and Sudan Red. Table 1 shows the IR vibrational modes of the three compounds. Although different, these compounds may have vibrational modes that are close or even in the same frequency range. Therefore, the assignment of vibrational frequencies to structural moities to is critical. Experimental evidence is needed to assign the spectral regions, and determine which intensities co-vary with concentration. The unique spectral region(s) with the highest relative intensity can be used as spectral fingerprints to identify the individual components in mixed samples. 

In Sudan Red (chemical structure shown in Figure 1b), the peak at 748 cm^−1^ assigned to bending of the N–C–C–O amide moiety (γ (N–C–C–O)_oop_ resonance V) is definitive [36]. Additionally, less intense peaks such as 835 cm^−1^ (out of ring plane C-OH bending [γ (C–OH)_ring_]), 868 cm^−1^ (ring breathing and stretching), 1107 cm^−1^ (aromatic CH bending in-plane), and 1205 cm^−1^ (CH_3_ deformation) are also confirmatory evidence of Sudan Red [37,38,39,40]. The vibrational mode in yellow turmeric at 1628 cm^−1^ assigned to the conjugated carbonyl group is not present in Sudan Red because the latter lacks this functional group [41,42]. The yellow turmeric peak at 1603 cm^−1^ due to C=C stretching is also fully discrete from Sudan Red. 

In white turmeric, the vibrational modes at 1078 cm^−1^ due to CH_3_ rocking and 1252 cm^−1^ due to CH bending are discrete and not present in yellow turmeric [43,44]. Furthermore, two peaks at 1381 cm^−1^ and 1628 cm^−1^ are markers for yellow tumeric but not for white tumeric, which means these are signature results of their spectral differences. The 1628 cm^−1^ versus 1635 cm^−1^ difference differentiates C=C stretching at sites in yellow tumeric having *trans*-orientation of methine hydrogens from the *cis*-orientation in white tumeric [41,42,45,46]. The second difference is that 1078 cm^−1^ in white tumeric is assigned to a site with a (non-terminal) methyl group rocking, whereas yellow tumeric has a mode at 1381 cm^−1^ assigned to OH bending at a CH_2_–OH methylene site [47].

### 3.2. Detection of Sudan Red Contamination in Yellow Turmeric

Figure 2a shows the deconvolved spectra of yellow turmeric (YT) with 1% Sudan Red (SR). The peaks at 748 cm^−1^, 835 cm^−1^, and 868 cm^−1^ indicate that SR can be detected at this concentration. The peaks at 1512 cm^−1^ and 1628 cm^−1^ are YT peaks. Most of the YT peaks are close to minor peaks of SR (Figure 1). The vibrational band pairs SR at 1020 cm^−1^ and YT at 1028 cm^−1^ (both ring breathing, 1149 cm^−1^ (Sudan Red CH_3_ rocking) and 1155 cm^−1^ (yellow turmeric CH_3_ rocking), 1381 cm^−1^ (yellow turmeric OH bending alcoholic), and 1385 cm^−1^ (Sudan Red Ar–O + Ar–N–R bending) are in a close frequency range. Due to the low intensity of the YT peaks, they were not resolved in mixed sample spectra.

Figure 2b shows the spectra of yellow turmeric—Sudan Red mixtures at 5% to 25% concentrations. Almost all the SR spectral peaks are identifiable at these higher levels, with certain marker peaks increasing in intensity with increasing SR concentration. Furthermore, certain YT peaks are suppressed as the SR concentration increases. For example, the YT peak at 1628 cm^−1^ identifiable in the 1% sample is not resolved in the 5% sample. In the 5% sample, the 1628 cm^−1^ YT peak and the 1616 cm^−1^ SR peak overlap *circa* 1616 cm^−1^. However, as the SR concentration increased further, 1616 cm^−1^ became resolved. Similarly, the YT peak at 1512 cm^−1^ was suppressed by the SR peak at 1500 cm^−1^. This indicates that the identified spectral peaks in Figure 3b are fingerprint peaks of Sudan Red. The regular pattern of increasing peak intensity also shows that the IR spectra can be used for quantitative analysis. 

PLSR is a widely accepted chemometrics method used to analyze spectral data for food authentication. However, the calibration models are prone to overfitting when a new data set is initiated. This is because the spectral data contain many factors, but only a few latent factors account for most of the variation. The PLSR method extracts the latent factors (PLS components) that are responsible for most of the variation, and uses these factors to develop the prediction models. The correct number of latent factors must be selected for accurate PLSR modeling. Selecting too many latent factors risks overfitting data. On the other hand, selecting too few latent factors decreases the overall accuracy of the model. The other method is the leave-one-out cross validation method where PLSR models (using different latent factors) are developed by removing one observation from the calibration set at a time until all the observations are removed at once. The RMSE is calculated, and the number of latent factors with the least RMSE is selected to develop the prediction model. This study used both methods to select the number of PLS components. 

Figure 3a shows the percentage variance vs. the number of PLS components. The variance increased steeply up to five components, and then flattened out. One PLS component has the least variance and 10 components shows the maximum variance. While the variance from five to 10 shows some increase, the difference is negligible (98.6% to 99.6%). To select the appropriate number of PLS components, models were developed by cross validation with PLS components from five to 10. Based on the least RMSE value, six PLS components were selected to develop the PLSR model. Using six principal components, the PLSR model resulted in R^2^_c_ = 0.98 and RMSEC = 0.9% in the calibration set. Figure 3b shows the residual errors of Sudan Red calibration set samples at 1% concentration. The errors are confined between −1% to +1%. This indicates that the PLSR model can be used to detect Sudan Red at a lower concentration. The model was used to estimate the Sudan Red concentration in the validation set. Figure 3c shows that the model yielded R^2^_v_ = 0.97 and RMSEP = 1.3%. The results show that FT-IR coupled with PLSR can be used for quantitative detection of Sudan Red contamination in yellow turmeric powder.

### 3.3. Detection of White Turmeric Adulteration in Yellow Turmeric

Figure 4a shows the spectra of yellow turmeric—white turmeric (YT-WT) mixtures at six concentration levels. The average spectra of each concentration are superimposed in Figure 4b. The peaks at 1078 cm^−1^, 1105 cm^−1^, 1512 cm^−1^, 1635 cm^−1^, and other minor peaks of both compounds, which were not readily identifiable in the original spectra and were enhanced after Fourier self-deconvolution. Most of the spectral peaks of both YT and WT are due to curcumin. Therefore, these are difficult to resolve. The 1628 cm^−1^ YT and 1635 cm^−1^ WT modes are in a close frequency range. These two peaks were not resolved and appeared as a single broad band at 1635 cm^−1^ in the mixed sample spectra. Similarly, the 1635 cm^−1^ WT peak and the 1512 cm^−1^ YT peak (Ar–O + Ar–O–R asymmetrical bending) were not resolved and appeared as a single band. The 1512 cm^−1^ curcumin peak intensity is expected to increase with greater curcumin content. YT has more curcumin content than WT does. Figure 4 shows that 1512 cm^−1^ increases in intensity as the WT concentration decreases from 50% to 1%. The 1512 cm^−1^ peak has the highest intensity at 1% adulterant (99% YT) and lowest at 50% (50% YT). The region *circa* 1105 cm^−1^ is due to overlapping minor peaks of white and yellow turmeric. Additionally, 1078 cm^−1^ is the unique spectral peak of WT: the pattern of increasing peak intensity with increasing WT concentration indicates this mode can be used to identify WT in the mixtures.

The IR spectra of YT-WT samples in the 900 to 1700 cm^−1^ range were used to develop a PLSR model. Based on the percentage variance vs. the number of PLS components (figure not shown) and leave-one-out cross validation method, eight PLS components were selected to develop the prediction model. Figure 5a shows the resultant model with R^2^_c_ = 0.97 and RMSEC = 2.8% in the calibration set. At 1% concentration, a large variation, which ranges from −5% to 15% in the predicted concentration, was observed. Figure 5b shows the residual error of WT calibration set samples at 1% concentration. The residual errors are scattered from −12% to 5%. This indicates that the model cannot be used to estimate WT at 1% concentration. The large discrepancy for prediction of the 1% WT sample is due to the similar spectral pattern of WT and YT.

WT can be added in YT to increase the product weight. Large amount of WT might be added in the YT powder. Mixture samples at higher concentrations (10%, 20%, 30%, 40%, and 50%) were used to develop a new PLSR model. A total of 150 samples were used to develop the calibration model (30 samples × 5 concentration range), and 100 samples (20 samples × 5 concentration range) were used to validate the model. The calibration model yielded R^2^_c_ = 0.96 and RMSEC = 2.5%. Figure 6 shows the validation result of the PLSR model with R^2^_v_ = 0.95 and RMSEP = 3.0%.

Deliberate profit-driven contamination and adulteration in dry food powders has become a major safety concern worldwide. Studies have shown that a high concentration of chemical contaminants are added to spice products. Mishra et al., (2007) [54] collected 800 chili samples from 16 states in India to screen and quantify Sudan dye contamination in chili powders. The study reported that 66% of the samples were contaminated by the dyes, among which the 90-percentile level of Sudan I content was 1.04% [54]. In this study, we used Sudan Red in the concentration range of 1% to 25% to develop a quantitative model to predict the concentration of the chemical contaminant in yellow turmeric-Sudan Red samples. While the precise concentration of white turmeric adulteration in yellow turmeric powder are not reported in scientific literature, a study by Sasikumar et al., (2004) [18] used three commercial brands of turmeric powder and found more white turmeric than yellow turmeric in all the three market samples. In this study, we used white turmeric in the concentration range of 10% to 50% to develop a quantitative model to predict the concentration of the botanical additive in yellow turmeric-white turmeric samples.

## 4. Conclusions

This study used FT-IR spectroscopy to detect Sudan Red and white turmeric (*Curcuma zedoaria*) adulteration in yellow turmeric (*Curcuma longa*). FT-IR spectra of all three powders were acquired and interpreted. Yellow turmeric and white turmeric exhibited similar spectra due to their curcumin content, whereas numerous major and minor peaks of Sudan Red were identified and served as spectral fingerprints. Yellow turmeric—white turmeric and yellow turmeric—Sudan Red samples were prepared at six concentration levels. Mixed powders were dissolved in methanol, which were extracted and dried to obtain well mixed samples. The unique peak at 1078 cm^−1^ was used to identify white turmeric. The most intense peak of Sudan Red at 748 cm^−1^ was used for its identification. A PLSR model for each yellow turmeric—Sudan Red and yellow turmeric—white turmeric sample was developed to estimate the adulterant concentrations. The results show that the PLSR model can estimate Sudan Red contamination in the concentration range of 1% to 25% with R^2^_v_ = 0.97 and RMSEP = 1.3%. Similarly, the model can estimate the white turmeric in the concentration range of 10% to 50% with R^2^_v_ = 0.95 and RMSEP = 3.0%. Both the models show high accuracy indicating FT-IR spectroscopy coupled with the PLSR model, which can be used for quantitative detection of botanical additives and chemical contaminants in turmeric.

## Figures and Tables

**Figure 1 foods-08-00143-f001:**
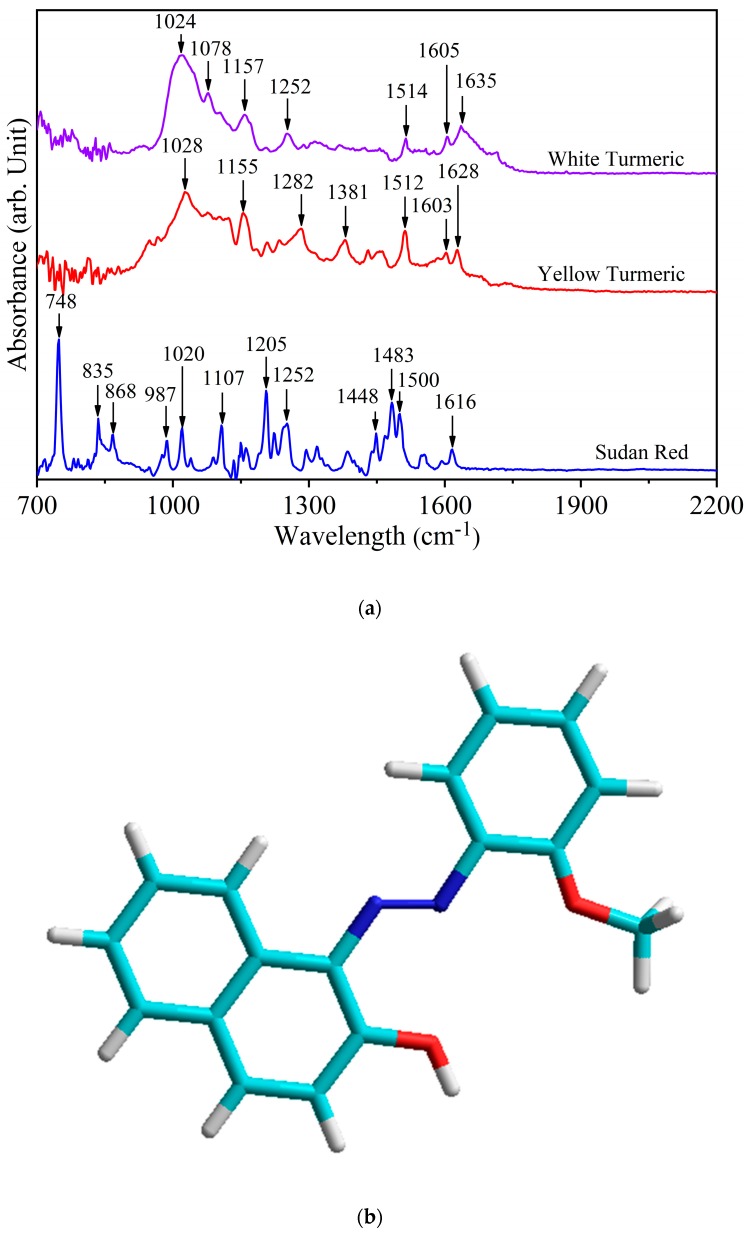
(**a**) FT-IR spectra of white turmeric, yellow turmeric, and Sudan Red. (**b**) Chemical structure of Sudan Red: light blue is carbon, dark blue is nitrogen, red is oxygen, and white is hydrogen.

**Figure 2 foods-08-00143-f002:**
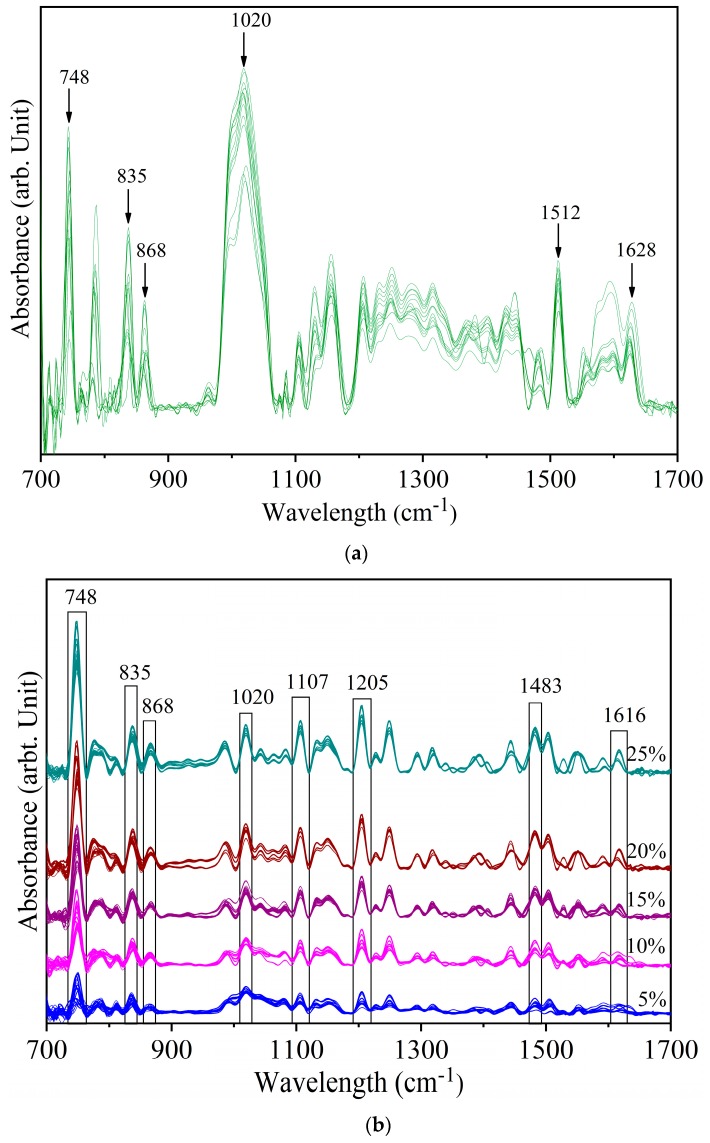
Spectra of yellow turmeric—Sudan Red sample (**a**) at 1% concentration (**b**) at 5%, 10%, 15%, 20%, and 25% concentrations.

**Figure 3 foods-08-00143-f003:**
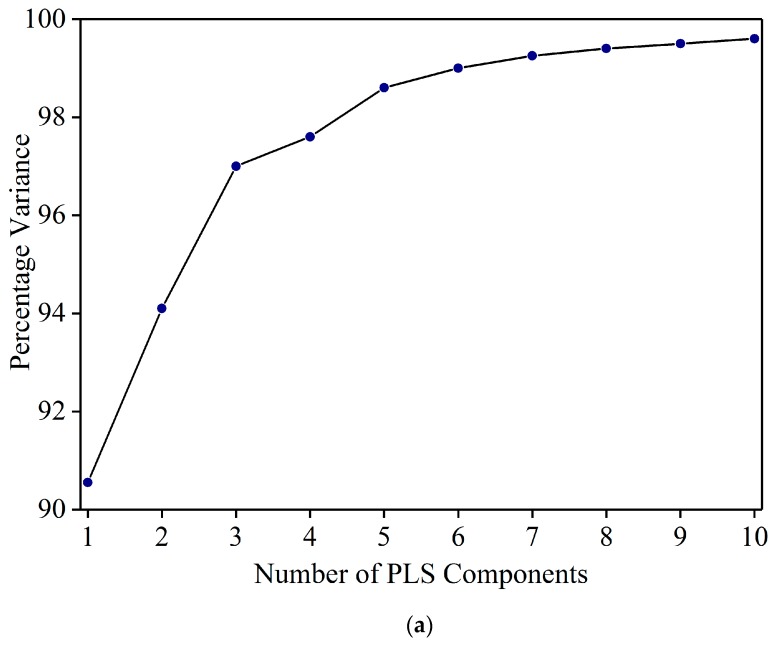
(**a**) Percentage variance explained by the number of PLS components. (**b**) Residual error of calibration set samples at 1% concentration. (**c**) Validation result.

**Figure 4 foods-08-00143-f004:**
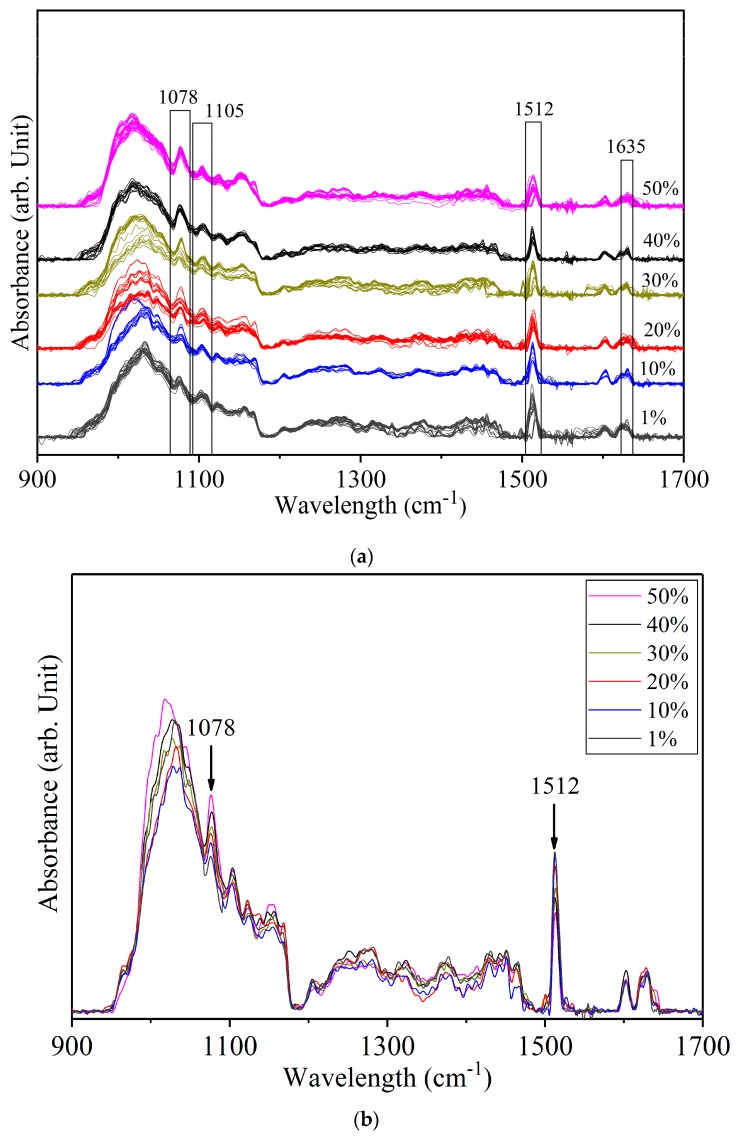
(**a**) Spectra of yellow turmeric—white turmeric samples at 1%, 10%, 20%, 30%, 40%, and 50% concentrations. (**b**) Average spectra of each concentration.

**Figure 5 foods-08-00143-f005:**
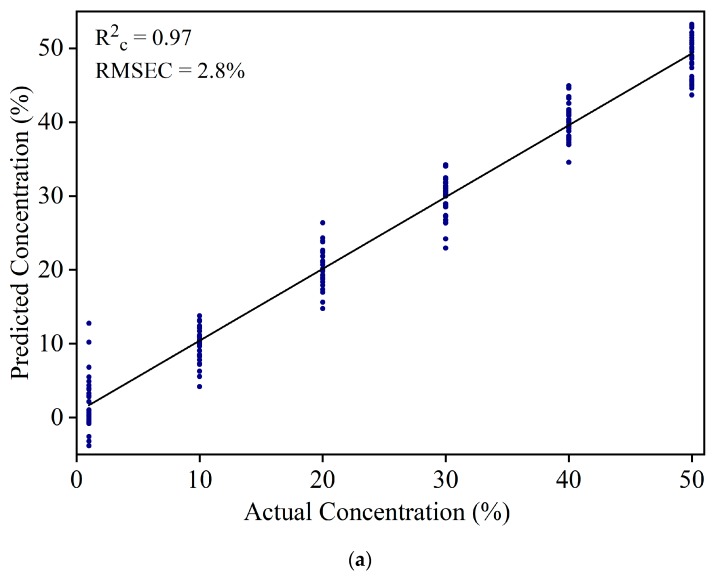
(**a**) Result of the calibration model to estimate white turmeric concentration. (**b**) The residual error for the calibration model of white turmeric at a 1% concentration.

**Figure 6 foods-08-00143-f006:**
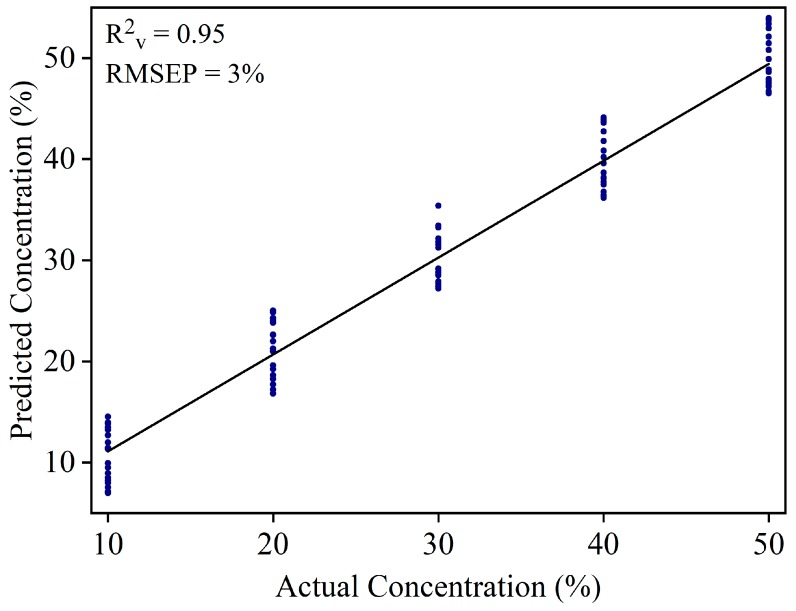
Validation result of the PLSR model of white turmeric samples at 10% to 50% concentrations.

**Table 1 foods-08-00143-t001:** Vibrational mode assignments for Sudan Red G, yellow turmeric, and white turmeric [36,37,38,39,40,41,42,43,44,45,46,47,48,49,50,51,52,53].

Sudan Red G	Yellow Turmeric	White Turmeric	Special Assignments
		1635	ν (C=O) stretching polycyclic quinones
	1628		ν (C=C) stretching *trans* form
1616			ν (C–N) stretching in C=C–N=N
	1603	1605	ν (C=C) stretching out of plane
1554			δ (N-H) bending in Ar–N=N··H–O–Ar’
	1512	1514	δ (Ar–O + Ar–O–R) bending asym
1500			ν (N=N) stretching *cis*
1483			ν (N=N) stretching *trans*
1448			ν (N=N) stretching + δ (H–C)
1385			δ (Ar–O + Ar–N–R) bending sym
	1381		δ (OH) bending alcoholic
		1369	δ (H–C) bending in O=C–CH_2_–C=O
1317			ν (Ar–O) stretching sym in Ar–N=N··H–O–Ar’
1294			ν (C–O)^−^ stretching on deprotonated Ar–O^−^
	1282		ν (C–O) stretching phenolic
1252		1252	CH bending
	1234		CH bending
1223			δ (O–H) bending asym
1205			CH_3_ deformation
1161			ν (C–N_azo_) δ(CH)
1149	1155	1157	δ (CH_3_) rocking, methoxy
1107			aromatic CH bending in-plane
		1078	ρ (CH_3_) rocking
1020	1028	1024	ring breathing
987			δ (C–N=N) out of plane bending (II)
868			ring breathing and stretching C–O
835			γ(C–O)_ring_ out of ring plane OH bending
748			γ(N–C–C–O)_oop_ resonance form of Amide V

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
