# Peer review of "Detection of Additives and Chemical Contaminants in Turmeric Powder Using FT-IR Spectroscopy"

_foods, 2019, doi:10.3390/foods8050143_

Round 1

Reviewer 1 Report

 This study used FT-IR spectroscopy for the identification and estimation of white turmeric (Curcuma zedoaria ), and Sudan Red G dye mixed with yellow turmeric powder. Thus, fifty replicate spectral measurements were obtained for each concentration, and partial least squares regression (PLSR) models were developed to estimate the adulterant concentrations.

In general, introduction must be improved giving additional references or adding information. For instance, in lines 43, 44, 48 and 49.

English language and style are correct. However, this methodology is not so novel since. there are a lot of previous studies reported in the literature whisky have been used FT-IR combined with PLSR to detect food frauds, but due to the type of samples, it could be of readers interest.

Author Response

We would like to thank the reviewer for providing valuable suggestion to improve the manuscript. All the changes are marked by blue color in the revised manuscript.

1. In general, introduction must be improved giving additional references or adding information. For instance, in lines 43, 44, 48 and 49.

Response: We have revised the manuscript with relevant information in the introduction.

2. English language and style are correct. However, this methodology is not so novel since. there are a lot of previous studies reported in the literature whisky have been used FT-IR combined with PLSR to detect food frauds, but due to the type of samples, it could be of readers interest.

Response: With the increasing popularity of turmeric powder, incidences of turmeric adulteration are also increasing. This study developed a simple method to quantify Sudan Red and white turmeric adulteration in yellow turmeric powder. The quantitative models developed in this study can be used to authenticate turmeric powder.

Reviewer 2 Report

It should be stated that the large part of this work, i.e. registration and assignment of FT-IR spectra of tumeric was previously reported by the same Authors. ( Foods 2016, 5, 36; doi:10.3390/foods5020036 )

line 155: turmeric is not a chemical compound so it is hard to tell about its "structure". It must be rewritten.

The chemical structure of Sudan Red should be provided.

line 239: Not deconvoluted but superimposed.

Figure 5: I am not convinced about the choice of the peak at 1078 cm-1. Could the Authors please provide the superposition of the spectra at the various concentration 1, 10, 20, 30, 40, and 50% (one of each) to inspect the visually?

Figure 6: I am not convinced about the practical application of the described methodology. Assuming the actual contamination at 0-5%, the predicted contamination is in wide range of -5 - +15%. I appreciate the easiness of this method, but it should be elaborated more.

The quality of figures, especially the axis descriptions, should be improved.

Author Response

We would like to thank the reviewer for providing valuable suggestion to improve the manuscript. All the changes are marked by blue color in the revised manuscript.

1. It should be stated that the large part of this work, i.e. registration and assignment of FT-IR spectra of turmeric was previously reported by the same Authors. ( Foods 2016, 5, 36; doi:10.3390/foods5020036 ).

Response:  Done. We have stated the registration and assignment of FT-IR spectra of turmeric was previously reported.  The following is added in the revised manuscript:

Dhakal et al. (2016) used Fourier Transfer-Infrared (FT-IR) and FT-Raman spectroscopic systems for qualitative detection of metanil yellow contamination in turmeric powder [2]. In the study, the vibrational modes of turmeric and metanil yellow were identified and the structural components were assigned to the respective vibrational frequencies. The highest spectral peak of metanil yellow at 1140 cm-1 was selected to detect metanil yellow in the mixture samples.

2. line 155: turmeric is not a chemical compound so it is hard to tell about its "structure". It must be rewritten.

Response: Yes, turmeric is not a chemical compound. We found the statement in line 155 is redundant. The statement is deleted in the revised manuscript.

3. The chemical structure of Sudan Red should be provided.

Response: The chemical structure of Sudan Red is shown in figure 1b.

4. line 239: Not deconvoluted but superimposed.

Response: The word is corrected.

5. Figure 5: I am not convinced about the choice of the peak at 1078 cm-1. Could the Authors please provide the superposition of the spectra at the various concentration 1, 10, 20, 30, 40, and 50% (one of each) to inspect the visually?

Response: In white turmeric spectra (figure 1a), the vibrational mode at 1078 cm-1 due to CH3 rocking is discrete- not present in yellow turmeric. The presence of 1078 cm-1 peak in the spectra of yellow turmeric-white turmeric mixture sample is a confirmatory evidence of the presence of white turmeric in the mixture samples.

Representative spectra of each concentration of WT-YT mixture samples (before applying Fourier deconvolution) are provided below for visual inspection. The 1078 cm-1 peak are distinct at 50%, 40% and 30% concentration samples. At 20% and 10% concentration, the peak is greatly reduced. The 1078 cm-1 peak is not readily identifiable in 1% sample spectrum. After applying Fourier deconvolution method (figure 4), the peak is distinct at all concentration range, indicating the peak can be used to identify white turmeric in the mixture samples. 

Figure - see attached PDF file.

6. Figure 6: I am not convinced about the practical application of the described methodology. Assuming the actual contamination at 0-5%, the predicted contamination is in wide range of -5 - +15%. I appreciate the easiness of this method, but it should be elaborated more.

Response: The PLSR model could not predict the white turmeric at 1% concentration. We re-analyzed the data and used white turmeric at concentrations 10, 20, 30, 40 and 50% to develop PLSR model. We have provided detail discussion about the practical application of the method in the line 289-323.

7. The quality of figures, especially the axis descriptions, should be improved.

Response: All the figures are revised to improve the quality of the axis descriptions.

Round 2

Reviewer 2 Report

The Authors have answered on some of my queries, yet some issues need to be solved.

The quality of Figure 1b is very poor, it looks like drawn in the MS paint. Please improve it.

I would like to see a Figure, which I suggest to put in the supplementary materials, in which the Spectra of yellow turmeric—white turmeric samples at 1, 10, 20, 30, 40, and 50% concentrations (one of each-total 6 spectra) should be superimposed.

Author Response

1. The quality of Figure 1b is very poor, it looks like drawn in the MS paint. Please improve it.

Response: We replaced figure 1b with an improved quality figure.

2. I would like to see a Figure, which I suggest to put in the supplementary materials, in which the Spectra of yellow turmeric—white turmeric samples at 1, 10, 20, 30, 40, and 50% concentrations (one of each-total 6 spectra) should be superimposed. 

Response: The superimposed spectra of yellow turmeric-white turmeric samples at each concentration are plotted in figure 4b of revised manuscript in page 10.

Round 3

Reviewer 2 Report

The Authors have answered on my questions.

Author Response

Thank you for the review and providing suggestion to improve our manuscript.